# SCALING 3D COMPOSITIONAL MODELS FOR ROBUST CLASSIFICATION AND POSE ESTIMATION

## ABSTRACT

Deep learning algorithms for object classification and 3D object pose estimation lack robustness to out-of-distribution factors such as synthetic stimuli, changes in weather conditions, and partial occlusion. Human vision, however, is typically much more robust to all these factors. This is arguably because human vision exploits 3D object representations which are invariant to most of these factors. Recently a class of 3D compositional models have been developed where objects are represented in terms of 3D meshes, with typically 1000 vertices associated with learnt vertex features. These models have shown robustness in small-scale settings, involving 10 or 12 objects, but it is unclear that they can be scaled up to 100s of object classes. The main problem is that their training involves supervised contrastive learning on the mesh vertices representing the objects and requires each vertex to be contrasted with all other vertices, which scales quadratically with the vertex number. A newly available dataset with 3D annotations for 188 object classes allows us to address this scaling challenge. We present a strategy which exploits the compositionality of the objects, i.e. the independence of the feature vectors of the vertices, which greatly reduces the training time while also improving the performance of the algorithms. We first refactor the per-vertex contrastive learning into contrasting within class and between classes. Then we propose a process that dynamically decouples the contrast between classes which are rarely confused, and enhances the contrast between the vertices of classes that are most confused. Our large-scale 3D compositional model not only achieves state-of-the-art performance on object classification and 3D pose estimation in a unified manner surpassing ViT and ResNet, but is also more robust to out-of-distribution testing including occlusion, weather conditions, and synthetic data. This paves the way for scalable 3D object understanding and opens exciting possibilities for applications in robotics, autonomous systems, and augmented reality.

## 1 INTRODUCTION

Recent progress in deep learning has yielded impressive results in different machine visual recognition tasks, such as object classification, detection, and pose estimation LeCun et al. (2015), with the help of large-scale training images and annotations. Cognitive scientists, however, suggest that human vision is more sophisticated and when classifying objects also recognizes their 3D structure including their shape and pose in a unified way using compositional representations Biederman (1987); Leek et al. (2005); Biederman (2000). We conjecture that endowing computer vision models with 3D representations will improve their performance, particularly in challenging out-of-distribution (OOD) scenarios, including domain shifts due to changes in weather, occlusions, and unfamiliar viewpoints, for which humans show big robustness Zhu et al. (2019), but where standard deep network models struggle Goodfellow et al. (2016); Koporec & Perš (2019); Zhu et al. (2019). The key insight is that the 3D structure of objects rarely varies in most OOD settings while deep network features are much more variable.

One promising avenue involves 3D Compositional Neural Networks (3D-CompNets) Wang et al. (2024; 2021); Ma et al. (2022); Jesslen et al. (2024); Kaushik et al. (2024b). These models are compositional in the sense that they represent objects by 3D meshes of vertices which are associated to learned vertex features which are independent both during learning and inference. The feature vectors are computed by a DNN feature extractor, CNN or Transformer, which is trained to encourage

them to be invariant to object viewpoint and instance. Using these compositional models, researchers demonstrated superior performance in generalizing to OOD scenarios for tasks such as image classificationJesslen et al. (2024), 3D pose estimation Wang et al. (2024); Kaushik et al. (2024b) and 6d pose estimation Ma et al. (2022). We note that the ability of these models to estimate object 3D pose, as well as independently, makes them suitable for a range of tasks such as robotic manipulation Jesslen et al. (2024)

However, to date, these neural mesh models have only been demonstrated on small datasets, such as Pascal-3D+ Xiang et al. (2014) (12 object classes) and OOD-CVZhao et al. (2024) (10 object classes), for two reasons. Firstly, because they require datasets with accurate 3D annotations for learning. Secondly, because their learning algorithms scale badly. For example, for previous neural mesh models Wang et al. (2024); Ma et al. (2022); Kaushik et al. (2024b); Jesslen et al. (2024), the contrastive learning includes every vertex from every object class which scales quadratically. The github code from Jesslen et al. (2024) will not run on more than 150 object classes due to memory limitations. This raises the challenges we address in this work: (I) Can 3D compositional models be scaled to a large number of object classes *efficiently*? (II) How will they perform compared to conventional neural networks in i.i.d. testing? (III) Most importantly, will they retain their important robustness properties (e.g., robustness to OOD, partial occlusion handling, multitasking) when scaled up?

In this work, we reformulate neural mesh models to allow scaling up to a large number (188) of object classes efficiently exploiting the recent availability of 3D annotated data Ma et al. (2023; 2024).

Our strategy is to train 3D compositional models by a new algorithm *compositional contrastive learning* that exploits the compositionality of our objects in terms of their vertices. We conjecture that only a small number of the huge number of contrastive vertex pairs are required to optimize the model to achieve strong performance. To identify these vertex pairs, our algorithm dynamically decouples the contrast between classes which are rarely confused and emphasizes the contrast between classes that are most confused. This greatly reduces the number of vertices of the object that need to be contrasted with, allowing for a greatly reduced computation. More precisely, we weight the contrastive loss between vertex features from a pair of object classes by their mutual confusion level (i.e. how similar the model considers the two object classes are). This is similar to classic hard-negative mining Xia et al. (2022); Kalantidis et al. (2020); Wang et al. (2023), but differs given that we exploit the compositional structure of our models in a supervised learning manner. In addition, to provide an improved rendering of image features, we modify the object mesh representations by using Gaussian splatting concepts so that a mesh vertex corresponds to a Gaussian with associated vertex feature.

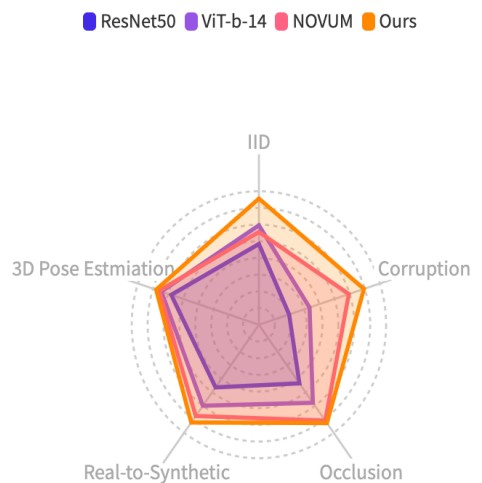

Figure 1: Our model is able to perform classification and 3D pose estimation simultaneously while being robust to IID and various OOD scenarios. (values are scaled for better visualization)

Concisely, our contributions are as follows:

1. We extend 3D-CompNets to an order of magnitude more object classes than previous studies and show they outperform conventional deep networks for both object classification and 3D pose estimation in a unified manner when tested on both IID and OOD data. By comparison, previous studies of 3D-CompNets showed no improvement over conventional deep networks on IID data.

2. We first refactor the per-vertex contrastive learning into two levels: contrasting within class and between classes to largely improve learning efficiency.

3. We further advanced the inter-class contrastive learning by dynamically decomposing object classes into subgroups and applying dynamic weights on contrasting between classes, enabling efficient and effective model optimization.

4. We further demonstrate that our model shows robust generalization capabilities from real-world to synthetic data, suggesting the potential to generalize from synthetic to real thereby mitigating the need for costly 3D annotations.

## 2 RELATED WORK

*Robust Image Classification and Pose Estimation* Deep Networks have been shown to be non-robust Schneider et al. (2020); Rusak et al. (2021); Kortylewski et al. (2021) to simple nuisances in tasks like image classification ima (2021); Hendrycks & Dietterich (2019); Hendrycks et al. (2021) and 3D pose estimation Zhao et al. (2024). Nuisances like partial occlusion, weather, additive noise, etc. may not have much effect on human visual capabilities however can completely derail deep neural networks outputs. A convincing theory attributes this fragility to lack of 3D compositional knowledge in these models which humans possess Kaushik et al. (2024b). Methods like data augmentations, test time adaptation, noise addition, input masking, etc. have been proposed to make neural models more robust with varying but unsatisfactory levels of success with many arguing that we would need a different architectural approach may be required Wang et al. (2017); Kortylewski et al. (2020) which incorporates some 3D object knowledge in the models.

*Robust Neural Compositional Models* It refers to a family of 2D Kortylewski et al. (2020); Kaushik et al. (2024a); Wang et al. (2017) and 3D models Wang et al. (2024); Jesslen et al. (2024); Kaushik et al. (2024b) who have shown to be robust to out-of-distribution nuisances like partial occlusion Kortylewski et al. (2020); Wang et al. (2021), real and synthetic corruptions Jesslen et al. (2024); Kaushik et al. (2024a;b) relative to conventional deep networks and have been utilized to perform robust image classification Kortylewski et al. (2020); Jesslen et al. (2024), 3D and 6D pose estimation Wang et al. (2024; 2021); Ma et al. (2022), amodal segmentation Sun et al. (2020) and unsupervised domain adaptation Kaushik et al. (2024a;b). These models focus on learning object-centric, compositional neural representations and often employ the ideas of analysis-by-synthesis Yuille & Kersten (2006) in their applications. However, all of these previous works have only been shown to work on small-scale datasets often due to the computationally expensive nature of learning these compositional, object-centric representations. In this work, we build upon ideas introduced by this family of models and scale them up efficiently to work with large datasets.

*Contrastive Learning.* Contrastive learning was originally developed for supervised learning Khosla et al. (2020); Misra & van der Maaten (2020); Wu et al. (2018); Chen et al. (2020); He et al. (2020) but has made its biggest impact when it was modified and applied to self-supervised learning Henaff (2020) giving state of the art results for many applications . Researchers have tried to adapt the idea of hard-negative mining Xia et al. (2022); Kalantidis et al. (2020); Wang et al. (2023) to improve performance and to improve efficiency but for unsupervised or self-supervised contrastive learning the lack of supervision makes it infeasible to adopt existing negative sampling strategies and motivates the development of other strategies Robinson et al. (2021). Although these strategies can be effective they are not always efficient because finding these hard samples takes time. Our approach differs in two respects. Firstly, it is supervised so it is easier to define a hard negative (e.g., two objects that are easily confused with each other). Secondly, we have a compositional structure of objects and parts and so we can use contrastive learning on the parts can be driven by hard negative mining of the objects.

## 3 METHOD : 3D COMPOSITIONAL NETWORK (3D-COMPNET)

In this section, we present a deep network architecture with an integrated object-centric 3D neural representation and an *efficient* learning algorithm (subsection 3.1) that can be used to perform robust image classification and 3D pose estimation *at scale* (subsection 3.2).

**Motivation and Problem Statement.** Previous methods Wang et al. (2024); Jesslen et al. (2024) learnt the *vertex features* by mapping the image feature at each 2D location from a feature extractor to a corresponding vertex in the 3D representation of the object given its 3D pose. The 3D representation

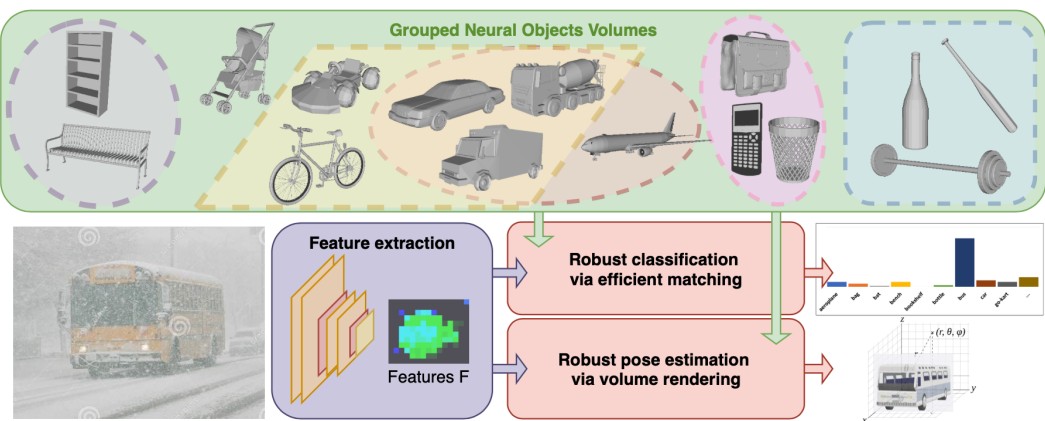

Figure 2: An overview of 3D-CompNet. The top-green box represents the variety of objects (188) that we consider and illustrates the grouping of Neural Vertex Features (NVF) (with arbitrary groups for illustrative purposes). The lower part illustrates the inference pipeline. During inference, an image is first processed by the backbone into a feature map $F$ (purple box). Then, by efficiently matching features from F and our NVF, the object class can be predicted (top-red box), or alternatively, given the class label, pose estimation can be performed by leveraging our volumetric representation in a render-and-compare manner (bottom-red box).

for each object class is either in a coarse shape like a cuboid or in an average prototypical shape. During training, the feature extractor is updated using *contrastive loss* between vertex features which ensures that every vertex feature is distinct from one another.

However, as we scale up to a large number of object classes $n$, we have to learn a large number of these compositional vertex representations. This is problematic for a few reasons:

1. We need to contrast every vertex feature with every other vertex feature of the same object class as well as all other object classes. The calculation/floating point operations grow with a complexity of $\mathcal{O}(n^2)$.

2. During training, model optimization (using the contrastive loss) becomes more complex as we increase the number of objects, due to the drastic increase in the number of vertex features.

3. During inference, we have to evaluate our data samples against the vertex features of all other classes and an incorrect classification inference may lead to incorrect pose inference.

### 3.1 GROUPED NEURAL OBJECT VOLUMES WITH DYNAMICALLY WEIGHTED COMPOSITIONAL CONTRASTIVE LEARNING

Expanding on recent advances in Gaussian splatting Kerbl et al. (2023), we articulate the representation of a set of objects as a three-dimensional density field through a spatial arrangement of $K$ Gaussians, strategically placed on the surface geometries of distinct object categories. Each Gaussian emanates a corresponding feature vector (which we will refer to as *volume features*), thereby delineating the volumetric representation termed as *neural vertex feature*. These features, representative of a single object category, is synthesized from feature maps derived from two-dimensional imagery, utilizing three-dimensional annotated poses of objects during training. For the task of classification and pose estimation, we find that cuboid geometries suffice Wang et al. (2021); Kaushik et al. (2024a); Jesslen et al. (2024) but more tightly defined geometries Wang et al. (2024) can also be used if available.

In a departure from previous works, we train vertice features in a *grouped* manner, what we refer to as **Grouped Neural Vertex Contrasting** using **Dynamically Weighted Compositional Contrastive Learning**. We learn neural object volumes (and their volume features) using a contrastive loss where the contrastive loss terms between similar object categories' volume features have a higher weight and dissimilar ones have a lower weight. This leads to a sparse and therefore more efficient contrastive loss calculation as most vertice feature pairs have the weight 0 associated with them i.e. we do not

calculate any corresponding loss terms. This contrastive loss formulation is termed *compositional* since every vertice feature is composed of individual volume features which roughly correspond to object parts. Additionally, we find that only a fraction of uniformly sampled volume features are necessary for the grouped contrastive loss calculation making the training process even more efficient.

Our grouped formulation helps us to ameliorate the drawbacks mentioned in the previous section 3. The **advantages** include

1. 95% reduction in the number of floating point operations for every contrastive loss calculation as we only calculate the distance between *uniformly-sampled* volume feature pairs of categories with *non-zero* weights.

2. Faster and easier contrastive loss optimization leading to better accuracy.

**Grouped Neural Vertex Contrasting**   Specifically, we define a neural volume density at spatial location $\boldsymbol{x} \in \mathbb{R}^3$ as a mixture of three-dimensional Gaussian $\rho_h(\boldsymbol{x}) = \sum_{k=1}^{K} \boldsymbol{\rho}_k(\boldsymbol{x})$. Each Gaussian density (what we refer to as *vertex feature*) $\rho_k(\boldsymbol{x})$ is defined as $\mathcal{N}(\boldsymbol{\mu_k}, \boldsymbol{\Sigma}_k)$, with $\boldsymbol{\mu_k} \in \mathbb{R}^3$ representing the position and $\boldsymbol{\Sigma}_k \in \mathbb{R}^{3 \times 3}$ the covariance matrix of the $k$-th vertex feature. The vertex features are arranged to form a cuboid volume with predefined diagonal covariance, covering the variable object instances in the category. Each vertex feature is linked to a feature vector $C_k \in \mathbb{R}^D$. We define the feature set for each category $y$ as $\mathcal{C}_y = \{C_k \in \mathbb{R}^D\}_{k=1}^{K}$, and the collective set across all categories and levels as $\mathcal{C} = \{\mathcal{C}_y\}_{y=1}^{Y}$, where $Y$ is the number of categories. The neural object volume can be rendered in the feature space, using standard volume rendering Jesslen et al. (2024):

$$\hat{\boldsymbol{C}}_i(\alpha) = \int_{t_n}^{t_f} T(t) \sum_{k=1}^{K} \rho_k(\boldsymbol{r}_\alpha(t)) \boldsymbol{C_k} \mathrm{d}t, \quad \text{where } T(t) = \exp\left(-\int_{t_n}^{t} \rho(\boldsymbol{r}_\alpha(s)) \mathrm{d}s\right), \quad (1)$$

where the feature $\hat{\boldsymbol{C}}_i(\alpha)$ at the pixel position $i$ in the rendered feature map is calculated by aggregating the vertex features along the ray $\boldsymbol{r}_\alpha(t)$. The ray passes through the centre of the camera through the pixel $i$ on the image plane with $\alpha$ denoting the camera view. Here, $t$ ranges from the near plane $t_n$ to the far plane $t_f$. The remainder of the image that is not covered by the rendered object volume is represented as background features $\mathcal{B} = \{\beta_n \in \mathbb{R}^D\}_{n=1}^{N_b}$ where $N_b$ is a fixed hyperparameter and $\mathcal{B}$ is shared across all categories. The background features are represented as von Mises-Fisher distributions with constant concentration parameters. We note that $\mathcal{B}$ can be replaced by a threshold Kaushik et al. (2024b) with similar performance to save computing resources.

Our model architecture builds on a feature extractor $\Phi_w$ and a set of neural volume features - each neural volume corresponding to one object category. The feature extractor $\Phi_w$, using CNN parameters $w$, transforms an input image $I$ into a feature map $F = \Phi_w(I) \in \mathbb{R}^{D \times H \times W}$. This map holds feature vectors $f_i \in \mathbb{R}^D$ at each 2D lattice position $i$.

Our model learns by mapping a vertex feature $k$ from a neural object volume to a location $i$ on the feature map $F$ of a training image, using the camera pose $\alpha$ and volume rendering to calculate the contribution $\gamma_{ik}$ of vertex feature $C_k$ to image features $f_i$. We establish a one-to-one correspondence by selecting the image feature $f_i$ closest to each vertex feature, specifically where $\gamma_{ik}$ is maximal. For clarity, $f_{k \to i}$ denotes the feature $f_i$ at location $i$ corresponding to vertex feature $k$ with mean $\mu_k$.

We model the probability of generating the feature $f_i$ from vertex feature $C_k$ as $P(f_{k \to i}|C_k) = c_M(\kappa)e^{\kappa f_{k \to i} \cdot C_k}$, with $C_k$ as the mean of each vMF distribution, both $f_{k \to i}$ and $C_k$ are unit vectors. Similarly, the probability of $f_i$ from background features $\beta_n$ is $P(f_i|\beta_n) = c_M(\kappa)e^{\kappa f_i \cdot \beta_n}$, where $\beta_n \in \mathcal{B}$. We define the concentration parameter $\kappa$, a measure of the spread of the distribution, as a global hyperparameter, allowing us to disregard the normalization constant $c_M(\kappa)$ during learning and inference.

**Dynamically Weighted Compositional Contrastive Learning**   Similarly to previous approaches Wang et al. (2021); Jesslen et al. (2024); Kaushik et al. (2024b), we maximize the probability that any extracted feature $f_{k \to i}$ was generated from $P(f_{k \to i}|C_k)$ instead of any other alternatives. This is done using a supervised contrastive learning formulation such that the likelihood that an extracted feature $f_{k \to i}$ is generated by the correct vertex feature $C_k$ is maximised Jesslen et al. (2024) w.r.t 1. from distanced vertex features of the same object 2. vertex features of other object classes,

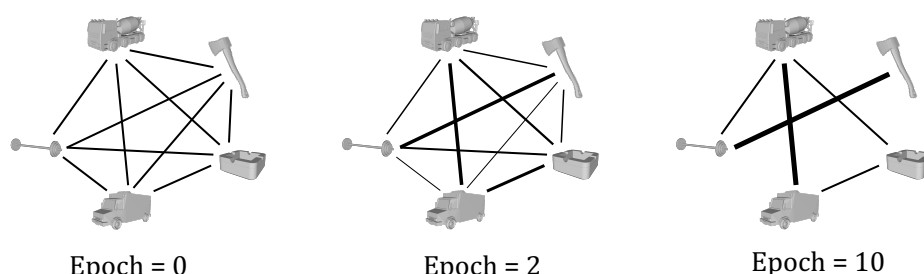

Figure 3: Illustrative example of our Dynamically Weighted Compositional Contrastive Learning. We change the weights applied on cross-category contrastive loss terms during training. The weights are calculated based on the confusion matrix from the calibration data split every two epochs.

and 3. background features. If we try to trivially scale this loss to $|Y|$ classes, we find that the number of contrastive terms scales approximately by a quadratic ($n^2$) factor! In addition, the loss landscape for optimizing over these many parameters further lengthens and complicates the training process. However, we *hypothesize* that all of these contrastive loss terms are not necessary and that we can make learning more effective by focusing on the most confused vertex feature pairs. We start training by unit-weighting every pair of vertex features. Note that to reduce the number of contrastive pairs, we uniformly sample only $30\%$ of the vertex features with which we contrast the current vertice features. Refer to ablation experiments in subsection 4.1. After every 2 epochs, we validate the model's performance on held-out calibration data. Using the confusion matrix from the calibration data split, we weigh the pairwise contrastive loss terms between 0 and 1. Weights are set to 0 when the confusion level is below 0.05 between object classes, means we don't calculate the contrastive loss between these classes anymore. This weighting changes dynamically throughout the training, and in the end will be sparse with the majority of the vertice feature pairs not being *grouped* together (i.e. 0 weight). Refer to Figure 3 for further insight.

We formulate the dynamically weighted compositional contrastive loss as follows -

$$\frac{P(f_{k\to i}|C_k)}{\sum_{\substack{C_l\in\mathcal{C}_y\\C_l\notin\mathcal{N}_k}} P(f_{k\to i}|C_l) + \omega_\beta \sum_{\beta_n\in\mathcal{B}} P(f_{k\to i}|\beta_n) + \Omega_{y,\bar{y}} \quad \omega_{\bar{y}} \sum_{\substack{\bar{C}_m\in\mathcal{C}_{\bar{y}}\\\bar{C}_m\sim U(\bar{C}_m)}} P(f_{k\to i}|\bar{C}_m)}, \quad (2)$$

where $\mathcal{N}_k = \{C_r : \|\boldsymbol{\mu}_k - \boldsymbol{\mu}_r\| < \delta, k \neq r\}$ is the neighborhood of $C_k$ and $\delta$ is a distance threshold controlling the size of neighborhood. $y$ is the category of the image and $\bar{y}$ is a set of all other categories except $y$. $\bar{C}_m \sim U(C_m)$ refers to uniformly sampled vertex features from Neural Object Volumes of all other categories. $\Omega_{y,\bar{y}}$ is the *grouping weight* which is calculated using the confusion matrix between object categories of the calibration dataset. The confusion matrix is normalized over the true (image class $y$), and a threshold of 0.05 is set to turn the grouping weight into 0 when the confusion level is below the threshold.

$\omega_\beta = \frac{P(\beta_n)}{P(C_k)}$ is the ratio of the probability that an image feature corresponds to the background instead of the vertex feature $k$, and $\omega_{\bar{y}} = \frac{P(C_m)}{P(C_k)}$ is the ratio of the probability that an image feature corresponds to vertex features of other categories instead of the vertex feature $k$.

We compute the final loss $\mathcal{L}(\mathcal{C}, \mathcal{B}, w)$ by taking the logarithm and summing over all training examples – all sets of features $\{f_{k\to i}\}$ from the training set

$$-\sum_y \sum_{k=1}^K o_k \cdot \log \frac{e^{\kappa f_{k\to i}\cdot C_k}}{\sum_{\substack{C_l\in\mathcal{C}_y\\C_l\notin\mathcal{N}_k}} e^{\kappa f_{k\to i}\cdot C_l} + \omega_\beta \sum_{\beta_n\in\mathcal{B}} e^{\kappa f_{k\to i}\cdot\beta_n} + \Omega y,\bar{y} \quad \omega_{\bar{y}} \sum_{\substack{C_m\in\mathcal{C}_{\bar{y}}\\\bar{C}_m\sim U(\bar{C}_m)}} e^{\kappa f_{k\to i}\cdot C_m}}, \quad (3)$$

where $o_k = 1$ if the vertex is visible and $o_k = 0$ otherwise and $y$ is the object category.

**Updating vertex features and Background Features.** The vertex features and background features $\mathcal{C}$ and $\mathcal{B}$ are updated after every gradient update of the feature extractor. Following He et al. (2020); Bai et al. (2023), we use momentum update for the vertex features:

$$C_k \leftarrow C_k \cdot \sigma + f_{k \rightarrow i} \cdot (1 - \sigma), \quad \|C_k\| = 1. \tag{4}$$

The background features are simply resampled from the newest batch of training images. In particular, we remove the oldest features in $\mathcal{B}$, i.e. $\mathcal{B} = \{\beta_n\}_{n=1}^N \setminus \{\beta_n\}_{n=1}^T$. Next, we sample $T$ new background features $f_b$ from the feature map, ensuring $f_b$ is not influenced by any vertex feature, and update $\mathcal{B}$ as $\mathcal{B} \leftarrow \mathcal{B} \cup \{f_b\}$. Note that $\sigma$ and $T$ are model hyperparameters.

## 3.2 Inference for Image Classification and 3D Pose Estimation

**Fast Robust Classification**  Image classification is performed swiftly and robustly by matching extracted features to learned vertex features of all vertex features and background features. For each category $y$, we compute both foreground $P(f_i|\mathcal{C}_y)$ and background $P(f_i|\mathcal{B})$ likelihoods across all feature map locations $i$. Ignoring object geometry simplifies this to a fast convolution operation. Image classification involves comparing average total likelihood scores across all locations for each class.

Specifically, we define a binary-valued parameter $z_{i,k}$ such that $z_{i,k} = 1$ if the feature vector $f_i$ matches best to any vertex feature $\{C_k\} \in \mathcal{C}_y$, and $z_{i,k} = 0$ if it matches best to a background feature. The object likelihood of the extracted feature map $F = \Phi_w(I)$ can then be computed as Jesslen et al. (2024):

$$\prod_{f_i \in F} P(f_i|z_{i,k}, y) = \prod_{f_i \in F} P(f_i|C_k)^{z_{i,k}} \prod_{f_i \in F} \max_{\beta_n \in \mathcal{B}} P(f_i|\beta_n)^{1-z_{i,k}}. \tag{5}$$

As described in subsection 3.1, the extracted features follow a vMF distribution. Thus the final prediction score of each object category $y$ is:

$$S_y = \sum_{f_i \in F} \max\{\max_{C_k \in \mathcal{C}_y} f_i \cdot C_k, \max_{\beta_n \in \mathcal{B}} f_i \cdot \beta_n\}. \tag{6}$$

The final category prediction is $\hat{y} = \arg\max_{y \in Y}\{S_y\}$.

**Volume Rendering for Pose Estimation.**  Given the predicted object category $\hat{y}$, we use the vertex feature $\mathcal{C}_{\hat{y}}$ to estimate the camera pose $\alpha$ leveraging the 3D geometrical information of the neural object volumes. Following the vMF distribution, we optimize the pose $\alpha$ via feature reconstruction Wang et al. (2024); Ma et al. (2022); Kaushik et al. (2024b); Jesslen et al. (2024):

$$\mathcal{L}(\alpha) = \sum_{f_i \in FG} f_i \cdot \hat{C}_i(\alpha) + \sum_{f_b \in BG} \max_{\beta_n \in \mathcal{B}} f_b \cdot \beta_n, \tag{7}$$

where $FG$ is the set of foreground features that are covered by the rendered neural object, i.e. those features for which the aggregated volume density is bigger than a threshold $FG = \{f_i \in F, \sum_{k=1}^K \rho_k(\mathbf{r}_\alpha(t)) > \theta\}$. $BG = F \setminus FG$ is the set of features in the background. Pose estimation begins by identifying the optimal initial pose $\alpha$ through computation of the reconstruction loss (Equation 7) across predefined poses. This is followed by gradient-based optimization starting from the pose with the lowest loss to determine the final pose $\hat{\alpha}$.

## 4 Experiments

In this section, we evaluate our approach and baselines on classification and 3D pose estimation tasks using both *synthetic* and real data in in-distribution and out-of-distribution (OOD) scenarios (subsection 4.1). Additionally, we show results on *large-scale synthetic-to-real data generalization* (subsection 4.3). Finally, we conduct ablation studies to analyze the key components (subsection 4.4). We included some qualitative results in Figure 7 of the appendix.

**Datasets** We use two different types of data in our experiments, notably real and synthetic data.
*Real Data* We train and evaluate our method on real data using the ImageNet3D dataset Ma et al. (2024), a large dataset for 3D understanding containing class and 6D pose annotation. From the

dataset, we selected a total of 188 classes with enough images for a total of 61 230 images divided in 30 630 training images and 30 600 test images. We then create occluded-Imagenet3D following Wang et al. (2020) by placing occlusion on both object and background in three levels: L1, L2, and L3. In L1, around 10% of the object and 30% of the background will be occluded, and these numbers are 30%, 50% for L2 and 50%, 70% for L3. We also test on corruptions following Hendrycks & Dietterich (2019) for 4 kinds of common types of corruptions in natural environment on level 4.

*Synthetic Data* For out-of-distribution testing, we also test our method on synthetic data generated following the approach outlined by Ma et al. (2023). This method enables precise 3D geometry control of diffusion models, allowing us to obtain detailed 3D annotations for the generated images. We generate the synthetic data for a subset of the object classes that exist in our real dataset. Hence, we have 50 synthetic classes and 500 images for each class. We included some visualisations of the generated synthetic data in Figure 6 of the appendix.

**Implementation** The features extractor $\Phi_w$ of our architecture builds is a ViT-B/14 with DI-NOv2 Oquab et al. (2023) pretraining. The input image size is $644 \times 812$ for ViT-B/14 backbone and the output feature map $F$ is $1/14^{th}$ of the input size. Output features are projected to a dimension of $D = 128$.

A baseline model with ResNet50 He et al. (2016) feature extractor has two upsampling layers to integrate the output from the last three layers of ResNet50. The size of the feature map $F$ is $1/8^{th}$ of the input size. Output features are projected to a dimension of $D = 128$.

Our method is trained as described in section 3.1. For each class, the corresponding vertex feature is composed of approximately $K = 600$ vertex features for each object class. To model the background, we use $N = 2560$ background features. We use momentum update for the vertex features using $\sigma = 0.9$ and sample $T = 5$ new background features from the image background to update $\mathcal{B}$ at each gradient step. Our model only takes 12 epochs to train fully.

**Evaluation** We evaluate all methods on two different tasks: image classification and 3D pose estimation. Image classification consists of estimating the object category of the main object in the image while 3D pose estimation involves predicting the azimuth, elevation, and in-plane rotation of an object to a camera. The pose estimation error is calculated between the predicted rotation matrix $R_{\mathrm{p}}$ and the ground truth rotation matrix $R_{\mathrm{gt}}$ as $e = \left\| \log m \left( R_{\mathrm{p}}^T R_{\mathrm{gt}} \right) \right\|_F / \sqrt{2}$, following Zhou et al. (2018). We define the coarse and fine accuracy of that task using two thresholds where a prediction is considered correct if $e < \frac{\pi}{6}$ and $e < \frac{\pi}{18}$.

**Baselines** We compare the performance of our approach to 2 competitive baseline architectures (that is, Resnet50 and ViT-b-16) for the classification and 3D pose estimation tasks. During training, these baselines are trained with a dual head: one for classification and one for pose estimation. This approach allows to leverage of the 3D knowledge for classification and conversely class knowledge for 3D pose estimation. For each baseline, the classification and pose estimation heads have an output size corresponding to the number of classes (i.e., 188) and the number of angles to estimate times the bin size (i.e., $3 \cdot 40 = 120$), respectively. We consider pose estimation as a classification task for each angle (ie elevation, in-place rotation, azimuth) by dividing the $360°$ pose space into 40 different classes of $9°$ each and predicting the corresponding bin. We finetuned each baseline for 100 epochs using standard cross-entropy loss, and chose the best checkpoint with the highest test accuracy on clean Imagenet3D+. In order to make baselines more robust, we apply standard data augmentation (i.e., scale, rotation, and flipping) during training.

### 4.1 Classification and 3D Pose Estimation

Table1, Table 3, and Table 4 show classification and 3D pose estimation performance on the base Imagenet3D dataset Ma et al. (2024), synthetic data generated using Ma et al. (2023), its corrupted version using corruptions like fog, snow, etc. from the Imagenet-C dataset Hendrycks & Dietterich (2019), and partial occlusion with levels ranging from $20 - 80\%$. All our baselines have 3D information incorporated in them during training. NOVUM Jesslen et al. (2024) is our ablative baseline, which is learned without *Grouped Neural Vertex Features* and without using *Dynamically Weighted Compositional Contrastive Learning*. All model performances reported here are trained till full convergence. In Table1, we show comparisons of classification task between our 3D-aware model and the same backbones with standard classification heads. Our model outperforms both

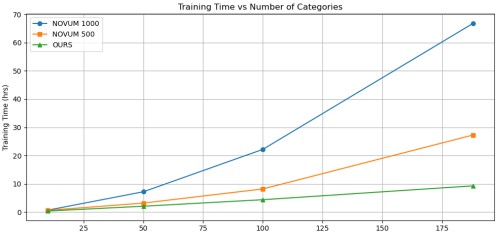

Figure 4: Training time comparison between NOVUM and our scalable 3D-CompNet

| Models | IID | Synthetic |
|---|---|---|
| Resnet50 | 84.8 | 58.2 |
| ViT-b-14 | 86.2 | 65.0 |
| NOVUM | 85.7 | 68.8 |
| OURS (ResNet50) | 86.5 | 69.3 |
| OURS (ViT-b-14) | **88.2** | **71.2** |

Table 1: Classification Results on Imagenet3D+

| NMMs | Neighbor size | Loss GFLOPS ↓ | Training Time ↓ | 12 Epoch | 100 Epoch |
|---|---|---|---|---|---|
| NOVUM | full | 61.2 (100%) | 66.8h | 30.0 | 85.7 |
| OURS | 64 | 4.31 (-93%) | 13.3h | 87.5 | - |
| OURS | 32 | **2.71 (-96%)** | **9.3h** | **88.2** | - |

Table 2: Detailed Comparison Between NOVUM and Our scalable 3D-compNet

the standard classification DNNs by 1.7% - 2.0% and the NOVUM baseline by 2.5% under the IID testing. Moreover, in Table 4, our model also shows the strongest performance on 3D pose estimation.

## 4.2 TRAINING TIME EFFICIENCY

We report quantitative results about the drastic decrease both in memory usage and the training time by our model in Table 2. Our model uses 96% less loss FLOPS and convergents 5 times faster than NOVUM, and still outperforms it and other standard neural networks thanks to our simple yet novel training methodology changes. Particularly, our model can converge with only 12 epochs of training, and we can outperform the performance of NOVUM being training 100 epochs to a final convergence.

Also, the training time of our model almost increases linearly with the number of categories, while for NOVUM, the number of computations scale quadratically ($O(n^2)$) with the number of categories. We compared our model with two NOVUM settings: NOVUM with 1000 vertices per mesh and with 500 vertices per mesh. We report the training time for each model to converge best with different number of categories in Figure 4. Considering more 3D data available in the future, an algorithm that scales up linearly is crucial both theoretically and practically.

| Classification under Occlusion and Corruption | | | | | | | | | |
|---|---|---|---|---|---|---|---|---|---|
| Model | L0 | L1 | L2 | L3 | **Average** | brightness | frost | snow | fog | **Average** |
| Resnet50 | 84.8 | 58.8 | 34.7 | 11.2 | 33.9 | 71.4 | 37.5 | 19.2 | 63.9 | 48.0 |
| ViT-b-14 | 86.2 | 60.1 | 35.9 | 12.9 | 36.3 | 68.3 | 48.3 | 21.8 | 64.3 | 50.7 |
| NOVUM | 85.7 | 64.6 | 37.6 | 13.4 | 38.5 | 75.1 | 46.1 | 30.1 | 72.6 | 55.9 |
| OURS | **88.2** | **65.2** | **37.8** | **13.5** | **38.8** | **78.8** | **48.6** | **30.4** | **73.6** | **57.9** |

Table 3: Classification results on clean, occluded, and corrupted ImageNet3D+. Different occlusion level (L1, L2, L3) and different corruption type applied.

## 4.3 DOMAIN SHIFT

Table 3 and Table 4 show that our method is able to outperform all other baselines with a large margin on both classification and 3D pose estimation with occlusion and corruption. We also report real-to-synthetic generalization performances for classification in Table1. We demonstrate our neural vertex features are strongly robust to various OOD scenarios under drastic domain shifts, including different levels of occlusion, unusual weather environments and domain shifts from real to synthetic.

| Pose Estimation under Occlusion and Corruption | | | | | | | | |
|---|---|---|---|---|---|---|---|---|
| Model | L0 | L1 | L2 | L3 | brightness | snow | frost | fog |
| Resnet50 | 55.6 | 40.4 | 27.5 | 14.4 | 50.8 | 29.1 | 38.7 | 51.3 |
| ViT-b-14 | 56.9 | 42.7 | 28.2 | 15.7 | 52.2 | 30.4 | 40.9 | 51.6 |
| NOVUM | 57.2 | 42.6 | 28.8 | 15.6 | 51.9 | 32.5 | 41.0 | 52.7 |
| OURS | **57.6** | **43.4** | **29.2** | **16.0** | **54.1** | **32.7** | **42.2** | **53.9** |

Table 4: 3D pose estimation on clean, occluded, and corrupted ImageNet3D+. Different occlusion levels (L1, L2, L3) and different corruption types applied. Results are estimated under accuracy $\pi/6$.

## 4.4 ERROR CASE ANALYSIS

Through per-category analysis on the IID performance, we found our 3D-compositional model performs well for most object classes, but not on elongated object classes, see Figure 5 for examples. The reason is that these objects look very similar, and sometimes even identical when facing forward and backwards, left and right, or when rotated along their dominant geometric axis. This ambiguity causes the main difficulty in estimating an accurate 3D pose of the objects. The 10 elongated objects are "ax", "paintbrush", "bow", "comb", "fork", "hammer", "french horn", "knife", "pen" and "pencil". Removing the elongated object classes from ImageNet3D+ leads to further improvement by our model, see Table 5.

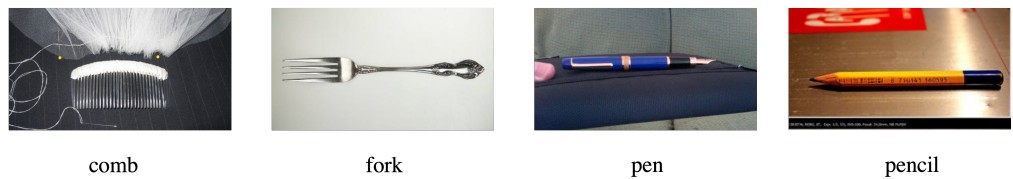

comb                fork                pen                pencil

Figure 5: Example images of elongated objects in the ImageNet3D+ dataset. For these ten classes, it is very difficult to estimate the 3D pose.

| Classification | | | | | 3D Pose Estimation | | | |
|---|---|---|---|---|---|---|---|---|
| | IID | Occ. | Corr. | | | IID | Occ. | Corr. |
| All classes | 88.2 | 38.8 | 57.9 | | All classes | 57.6 | 29.5 | 45.7 |
| w/o Elongated | **89.3** | **39.7** | **58.5** | | w/o Elongated | **59.3** | **32.8** | **48.3** |

Table 5: Comparison on the classification and pose estimation results by our model on the object classes including and excluding the ten elongated objects in ImageNet3D+. Here Occlusion (Occ.) and Corruption (Corr.) results are averaged.

## 5 DISCUSSION AND CONCLUSION

In this work, we argue that endowing computer vision object models with 3D representations will improve their performance, particularly in challenging out-of-distribution (OOD) scenarios. To demonstrate this, we scaled up 3D-CompNets to 188 object categories taking advantage of a recent dataset with 3D ground truth annotation. This scaling up required developing a modification of supervised contrastive learning, called *Grouped neural Vertex with Dynamically weighted Compositional contrastive Learning*(GVDComp). This algorithm exploited the compositional structure of 3D-CompNets and can be used for other applications of contrastive learning of compositional models. GVDComp resulted in greatly increased speed and reduction in computation, for learning 3D-CGNs, which increase significantly with the number of object classes. The resulting 3D-CompNets not only outperformed conventional neural networks for object classification and 3D pose estimation when tested IID (unlike 3D-CompNets tested previously) but also outperformed them even more significantly when tested in challenging OOD settings. We conclude that endowing object models with explicit 3D representations has many advantages including improved performance and robustness, as demonstrated in this paper, as well as applicability to a greater range of tasks including robotic manipulation.

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

# A APPENDIX

## A.1 SYNTHETIC DATASET VISUALISATION

In order to evaluate our method in many different settings, we generated 3D consistent data following Ma et al. (2023). Given some 3D CAD models, we were able to generate data with known objects class an 3D pose annotation. The usage of synthetic data is appealing since it allows to control many parameters during the dataset generation. Benchmark datasets like ImageNet3D can have certain bias (e.g., imbalance in the number of objects per class). Hence, we decided to generate synthetic images to measure our model's capacity to adapt to domain shift (i.e., real-to-synthetic generalization). In order to show the quality of the generated images, we show a subset of the generated data in Figure 6.

## A.2 QUALITATIVE RESULTS

In Figure 7, we provide a few qualitative results. We provide an example for the clean images of ImageNet3D, an example of synthetic occlusion of occluded-ImageNet3D, and two examples of corrupted images (notably *fog* and *pixelate*). We represent side-by-side the input image along with the input image overlaid by the prediction of our approach. We selected the CAD model of the class that was predicted by our approach and we overlaid the CAD model in the pose predicted by our approach.

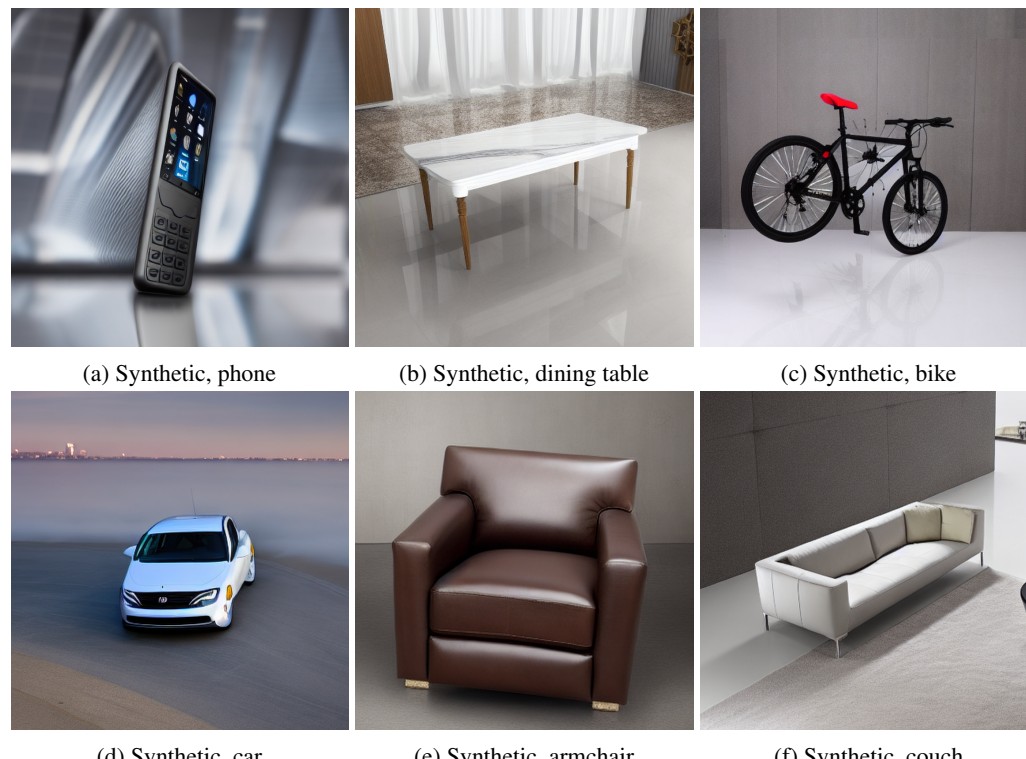

(a) Synthetic, phone  (b) Synthetic, dining table  (c) Synthetic, bike

(d) Synthetic, car  (e) Synthetic, armchair  (f) Synthetic, couch

Figure 6: Visualisation of the generated synthetic data.

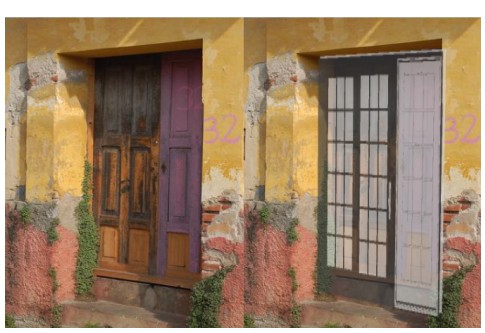
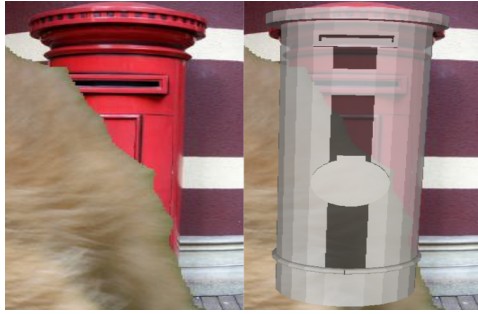

(a) Clean image, category: door  (b) Occ L2 image, category: fire extinguisher

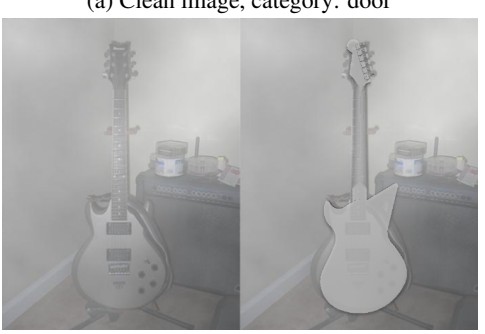
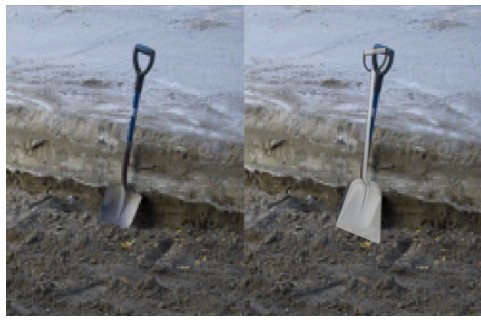

(c) Fog corrupted image, category: guitar  (d) Pixelate corrupted image, category: shovel

Figure 7: Qualitative results showing the predictions of our approach for classification and 3D pose estimation

