# OpenReview forum: "Scaling 3D Compositional Models for Robust Classification and Pose Estimation"
_ICLR.cc/2025/Conference — ICLR 2025 Conference Withdrawn Submission_

### Official Review · Reviewer_MovF · 2024-10-28

**Soundness:** 2
**Presentation:** 2
**Contribution:** 2
**Rating:** 3
**Confidence:** 4

**Summary:**

This paper solves the task of joint object classification and object rotation estimation, for both the in-domain and out-of-distribution settings. The methods is built on top of existing work NOVUM, but the authors propose two improvements. Firstly, the contrastive learning objective is breaked down into a within-class and a between-classes term. Secondly, Dynamically Weighted Compositional Contrastive Learning is proposed, where the object classes are divided into subgroups and the connections among classes are gradually pruned. The authors train their method on the large-scale realworld dataset ImageNet3D, and show an improvement in accuracy and a reduction in training time.

**Strengths:**

1. The proposed dynamic weights contrastive learning is interesting and well suited for the task involving many classes.
2. This paper uses ImageNet3D for training, which is much larger in scale compared to previous works. This is towards a good direction of scaling up the model and its generalization ability.

**Weaknesses:**

1. I'm concerned about the limited improvement over the baselines, given that the model is much more complicated than the ViT/ResNet baseline. For example, as shown in Tab 4, pose acc=57.6 for proposed method vs 56.9 for ViT on L0 occlusion, and 16.0 vs 15.7 for L3. One advantage is training time reduction, but I find this less important than the inference speed, as training only need to be done once.
2. I'm not sure if the comparison is fair, as some improtant experiment details are missing: L376-383 describes how virtual corrption on data is applied. Is this corrpution only applied to test set or also the training set? If only applied to test set, what data augmentation is used during training? If applied also to training, then this is different from the augmentation used to train baselines (we apply standard data augmentation (i.e., scale, rotation, and flipping) during training, quoted from L419-420), and the comparsion is not fair. The fairness of comparison is especially important given that the improvement is very small. Please clarify.
3. Fig 1 is misleading: the is actually not as big as shown in the figure. The authors should show the ticks on each axis.
4. False claim #1: In abstact and intro, the authors mention heavily their method can scale to higher number of vertices in the mesh. However, the main contribution of this paper (Dynamically Weighted Compositional Contrastive Learning) solves mainly the scaling to increasing number of classes. Neither is the generalization to higher number of vertices supported by any experiment. Therefore, I don't think this is a valid point.
5. False claim #2: L457 the authors say "...about the drastic decrease both in memory usage...by our model in Table 2.". However, there is no numbers for memory usage in Tab 2. Conceptually I also don't think the proposed model will have advantage in peak memory consumption, as the proposed dynamically weighted graph is pruned gradually, so dense connection is required at the beginning of the training.
6. False claim #3: L151 the authors claim "We have a compositional structure of objects and parts and so we can use contrastive learning on the parts". However, the object is represented as cubiods as a whole and no parts are used in their method.
7. I'm concerned about the real-world application of this method. The authors show that the method can genealize from real-world data to the diffusion model generated synthetic image, but it would be more convincing if the authors do the opposite, i.e., sim to real. Moreover, the pose estimation task is 3D instead of 6D. That means the object has to be at a fixed scale and centered at the location of the image. Also, it can only estimate pose of a fixed class of objects, since classification happens before pose estimation, which seems more limited than the stadard 6D pose estimation setup. The authors should at least present some qualitative results on in the wild images.
8. Conceptually, why is the proposed model more robust to occlusions for pose estimation? In Eqn 7, the authors maximize the feature similarity of both foreground region (defined by the rendered object mask) and background region. This foreground mask doesn't model occlusion, therefore the occluder (which has image feature corresponding to background) will be matched with object features. I think this will cause confusion in the pose etimation process. Some further clarification and discussion are desired.
9. L292 mentions there is an ablation for subsampling the vertices in contrastive loss but I don't find the table. Which exact ablation experiment should I look at?
10. The paper doesn't discuss about the failure cases or limitations. Also, more visualizations are desired to better understand how the leanred object representation looks like and how the pose estimation is performed.

Some minor problems that won't affect my rating:
M.1. Eqn 7, from my understanding this term should be maximized since you want to have a high matching score. In this case I would not call it a loss, as in ML loss is usually supposed to be minimized. Maybe name it "energy" instead.
M.2. Some abbreviations used without definition, and are inconsistent. E.g., two forms of IID used ("i.i.d" in L69 and IID in L103) without definition. O(n^2) is used in many places, but n is not defined.

**Questions:**

1. About the object representation: what are the optimizable parameters for the object representation? Are the location of the 3d gaussians in the cuboid learnable? How are the mean and covariance initialized? Compare Eqn 1 to the standard 3DGS, your model has no learnable opacity, why is it designed that way?

---

### Official Review · Reviewer_JF6D · 2024-11-01

**Soundness:** 2
**Presentation:** 1
**Contribution:** 2
**Rating:** 5
**Confidence:** 2

**Summary:**

This study scales up 3D-CompNets to more object categories by introducing 3D Gaussian representation and designs a Grouped neural Vertex with Dynamically weighted Compositional contrastive Learning (GVDComp) strategy, which can dynamically decouple the contrast between classes rarely confused and emphasize the contrast between classes most confused. The extended 3D-CompNets outperform conventional deep networks for both image object classification and camera pose estimation when tested on both IID and OOD data.

**Strengths:**

(1) The dynamic weights on compositional contrastive learning is plain but make sense. It can be used well on efficient and effective model optimization.

(2) Leveraging 3D Gaussian representation to solve both image object classification and camera pose estimation in a unified manner is interesting and reasonable.

(3) The extended 3D-CompNets scales up the number of the object categories and outperforms on both IID and OOD data.

**Weaknesses:**

1. There is still room for improvement in the writing of this paper. For me, it takes some time to understand the logic in the method section, especially since it contains a lot of unofficial expressions. Take a few examples:

a) line 269: w.r.t 1. from distanced vertex features of the same object 2. vertex features of other object classes and 3. background features.
（The three points listed lack punctuation, and I am unsure if this listing format is formal.）

b) line 286: If we try to trivially scale this loss to |Y | classes, we find that the number of contrastive terms scales approximately by a quadratic (n2) factor! (Informal)

2. There are also some unclear parts in the writing. For example, how the confusion matrix is computed from the calibration data split (line 293) and how the candidate poses set which is used to select the optimal initial pose α are predefined (line 366).

**Questions:**

1. When making evaluation on camera pose estimation, the authors optimize the optimal initial pose selected from a set of predefined poses on their extended 3D-CompNets. But on the 2 baseline (Resnet50 and ViT-b-16), I am not sure that it is fair to consider pose estimation as a classification task? Predicting the angle bin is kind like to select the best initial pose, what if author utilize a pose regression module to search the local region around the angle bin?

2. Although the method currently demonstrates superior capabilities compared to CNNs, it is trained on over a hundred classes. As I understand, image data and categories can currently scale up to a large extent. If in the future, the scale of 3D data allows for it, would this method also be able to operate on a larger set of categories?

---

### Official Review · Reviewer_CSrS · 2024-11-04

**Soundness:** 1
**Presentation:** 3
**Contribution:** 1
**Rating:** 5
**Confidence:** 4

**Summary:**

The paper proposes 3D-CompNet for class and pose estimation of the in-the-wild images. The paper proposes taking DiNov2 features with a neural field update and dynamically weighted contrastive loss in particular. Experiments on the ImageNet3D dataset and other dataset show the effectiveness of the approach.

**Strengths:**

+ The idea of scaling up the class and pose-estimation is nice.
+ Dynamically weighted contrastive loss to avoid too many comparisons makes sense.
+ Experiments show the effectiveness of the approach.

**Weaknesses:**

- The paper does not show results on the Omni3D [A] benchmark: a dataset comprised of 6 datasets. It would be good to quantitatively compare agains the Cube R-CNN [A] baseline on the pose estimation task on this task. Since Omni3D does not have its own pose estimation metric, you could use your metric to compare against the Cube R-CNN baseline on the pose estimation sub-task.

- What is the inference time of this method compared to Cube R-CNN baseline?

- Why does the method learn good neural volumes even if the number of input images is only one. Neural volumes are poorly constrained for 1 input view in my experience.

Reference:
- [A] Omni3D: A Large Benchmark and Model for 3D Object Detection in the Wild, Brazil et al, CVPR 2023

**Questions:**

Please see the weakness.

---

### Official Review · Reviewer_1NZw · 2024-11-04

**Soundness:** 3
**Presentation:** 3
**Contribution:** 3
**Rating:** 6
**Confidence:** 4

**Summary:**

The paper proposes a scheme for scaling the training time of 3D compositional models for 3D object classification and 3D pose estimation. The proposed scheme is based on the assumption of independence of feature vectors at distinct 3D mesh vertices. This assumption is exploited to reduce the training time by refactoring the per-vertex contrastive learning into contrasting within class and contrasting between classes. The contrastive learning is decoupled to enhance the contrast between classes that are most confused compared to the contrast between classes that are rarely confused.

**Strengths:**

The paper addresses an important problem of scalability of compositional model learning. The paper is technically sound and well presented. The experimental results are encouraging when compared to the state-of-the-art.

**Weaknesses:**

The assumption of the independence of feature vectors at each mesh vertex is a strong assumption. The authors need to come up with a strong justification for this assumption especially if one considers the spatial coherence of mesh vertices, i.e., the vertices that are in close spatial proximity would be expected to have similar feature vectors. Also, is the contrastive learning technique presented in the paper adequate to enforce the independence of feature vectors at distinct mesh vertices? Also, it is not clear how the training would scale with updates to the classes? Would the training need to be performed from scratch with the addition of a new class?

**Questions:**

How does one ensure the independence of feature vectors at distinct mesh vertices?
Have the authors considered techniques such as independent component analysis (ICA) as an alternative to the contrastive learning technique presented in the paper to ensure the independence of feature vectors at distinct mesh vertices?
How does one ensure scalability of the training procedure as new classes are added?

---

### Note · Authors · 2024-11-15

**Comment:**

After carefully reviewing the comments, we’ve decided to withdraw the paper to focus on further developing and refining our idea. The feedback has been invaluable, and we sincerely appreciate the reviewers’ time and effort. Thank you for your support in helping us improve this work.

**Withdrawal Confirmation:**

I have read and agree with the venue's withdrawal policy on behalf of myself and my co-authors.